# Medium Internal Phase Emulsions Stabilized by Soy Protein Isolates: Protein Solubility Effect and Stabilization Mechanism

**DOI:** 10.3390/foods14122028

**Published:** 2025-06-08

**Authors:** Fengxian Guo, Yiming Mao, Yujie Chen, Shiying Wu, Zhiyong He, Baobei Wang, Hongbin Chen, Shunhong Wu, Zongping Zheng

**Affiliations:** 1Fujian Province Key Laboratory for Development of Bioactive Material from Marine Algae, College of Oceanology and Food Science, Quanzhou Normal University, Quanzhou 362000, China; guofx0608@qztc.edu.cn (F.G.); maoyiming225@163.com (Y.M.); 19945263030@163.com (Y.C.); wying5533@163.com (S.W.); baobeiw@qztc.edu.cn (B.W.); yummyway@163.com (H.C.); 23064@qztc.edu.cn (S.W.); 2College of Material Engineering, Fujian Agriculture and Forestry University, Fuzhou 350108, China; 3State Key Laboratory of Food Science and Technology, Jiangnan University, Wuxi 214122, China; zyhe@jiangnan.edu.cn

**Keywords:** solubility, soybean isolate proteins, medium internal phase emulsions, microstructural analysis, protein adsorption

## Abstract

The solubility of soybean isolate protein (SPI) undergoes significant degradation during storage and transportation. This study investigates the formulation and assessment of SPI-stabilized medium internal phase emulsions (MIPEs) with different solubilities, namely SPI80, SPI70, SPI60, and SPI50, corresponding to solubility levels of about 80%, 70%, 60%, and 50%, respectively. The contact angles of these SPI variants ranged from 79.35 to 86.55 degrees, with SPI60 and SPI50 exhibiting significantly higher values compared to SPI80 and SPI70. All SPI samples were successfully utilized for the preparation of MIPEs. However, as SPI solubility decreases, emulsion stability progressively declines, accompanied by a reduction in the absolute value of zeta potential. Additionally, interfacial protein adsorption in emulsions decreases with decreasing SPI solubility, a trend that is similarly observed in viscosity characteristics, storage modulus (G′), and loss modulus (G″). Confocal laser scanning microscopy (CLSM) and cryo-scanning electron microscopy (Cryo-SEM) analyses revealed that emulsions exhibit reduced uniformity and a less interconnected microstructural network as SPI solubility decreases. These findings provide a theoretical foundation for utilizing low-solubility SPI in MIPEs applications.

## 1. Introduction

Soy protein isolate (SPI) is among the most widely commercialized plant-based protein products due to its desirable functional properties and high nutritional value. Furthermore, its abundant availability and cost-effectiveness have contributed to its recognition as a sustainable raw material [1]. However, variations in environmental factors, including temperature and pH fluctuations during storage and transportation, lead to a significant decline in protein solubility, thereby restricting its applications and reducing its commercial value [2]. Existing studies on low-solubility soy proteins primarily focus on methods to restore solubility [3] through processing techniques such as ultrasound treatment [4] and pH adjustment [5]. However, these additional processing steps increase production costs and pose challenges for industrial applications.

Emulsions are widely utilized multiphase dispersion systems consisting of two or more immiscible liquids. In recent years, increasing attention has been directed toward the application of soy proteins in emulsion systems [6]. For example, SPI has been employed as a stabilizing agent in various emulsions, including high internal phase emulsions (HIPEs) with internal phase volume exceeding 74% [7], medium internal phase emulsions (MIPEs) with internal phase volume around 30–74% [8], emulsion gels [9], emulsion membranes [10], and oleogels [11]. Theoretically, compared to HIPEs, MIPEs can better meet consumers’ health needs and are more widely applied in food systems, such as ice cream [12,13], salad dressing [14], and other products. However, previous studies have primarily focused on the properties and behavior of highly soluble SPI.

The influence of insoluble proteins on the functional properties of emulsions, particularly in relation to soy protein isolate (SPI), remains relatively underexplored. Insoluble proteins can have a significant impact on emulsion characteristics. Previous studies have demonstrated that insoluble particles, such as starch [15], corn protein [16], and walnut protein [17], play a crucial role in emulsion stabilization. Upon adsorption at the liquid–liquid interface, these particles form a robust physical barrier that prevents droplet aggregation and coalescence, thereby enhancing emulsion stability and reducing phase separation [18]. In emulsion systems containing both soluble proteins and insoluble particles, stabilization effects have shown inconsistent outcomes. For instance, insoluble pea protein has been identified as an effective particle stabilizer, contributing to the formation of highly stable high internal phase emulsions with increased viscosity [19]. Additionally, soluble SPI complexed with chitosan nanoparticles has been reported to exhibit enhanced emulsification properties [20]. These findings indicate that in composite systems, solid particles primarily stabilize emulsions by forming a physical barrier at the water–oil interface, while soluble proteins further improve interfacial stability through electrostatic interactions. However, studies have also shown that fava bean isolate proteins demonstrate a reduced emulsification capacity when solubility decreases [21].

This study aims to evaluate the performance of soy protein isolate (SPI) with varying solubility in medium internal phase emulsions (MIPEs), providing both theoretical and practical insights into the application of low-solubility SPI in MIPEs. The investigation systematically examines the influence of SPI solubility on MIPE stability, particle size, ζ-potential, interfacial protein adsorption rate, and rheological properties. Furthermore, the microscopic morphology of different MIPEs was analyzed using laser confocal microscopy and cryo-scanning electron microscopy to elucidate the relationship between SPI solubility and MIPE characteristics. By comprehensively assessing the behavior of SPI with different solubility levels in MIPEs, this study establishes a scientific foundation for the development of low-solubility SPI MIPEs and explores novel strategies for improving SPI functionality.

## 2. Materials and Methods

### 2.1. Materials and Chemicals

Low-temperature defatted soybean meal was obtained from Shandong Yuwang Industrial Co., Ltd. (Dezhou, China). Bovine Serum Albumin (BSA) was procured from Hefei Bomei Biotechnology Co., Ltd. (Hefei, China) Soybean oil (Jinlongyu) was sourced from local supermarkets. Nile Red, Nile Blue, 8-Aniline-1-naphthalenesulfonic acid (ANS), and Sodium Hexadecyl Sulfate (SDS) were supplied by Shanghai McLean Biochemical Science and Technology Co., Ltd. (Shanghai, China). All other reagents utilized in this study were of analytical grade.

### 2.2. SPI Preparation

SPI was extracted following the method of Ju [22] with modifications. Defatted soybean flour was mixed with deionized water at a solid–liquid ratio of 1:6, stirred for 1 h at pH 8.0, and subsequently centrifuged (GL-21M, Hunan Xiangyi Laboratory Instrument Development Co., Ltd., Changsha, China) at 10,000× *g* for 25 min at 20 °C. The resulting supernatant was adjusted to pH 4.5 and centrifuged at 3000× *g* for 10 min at 20 °C to obtain the protein precipitate. The precipitate was washed with deionized water, followed by centrifugation (3000× *g*, 10 min, 20 °C), then dissolved in deionized water and adjusted to pH 7.0. Protein concentration was determined using the Biuret method [23], with BSA as the standard protein, and subsequently diluted to 5% (*w*/*v*) with deionized water.

SPI powders with varying protein solubility were prepared following the method of Guo et al. [24]. A diluted SPI suspension (5%, *w*/*v*) was subjected to heat treatment at 80 °C for 5 min, 90 °C for 5 min, and 90 °C for 15 min, followed by rapid cooling in an ice bath. The heat-treated SPI suspensions were then spray-dried using a spray-drying machine (ADL311-A; Yamato Holdings Co., Ltd., Yamanashi, Japan) with an inlet temperature of 180 °C, an outlet temperature of 90 °C, at 0.1 MPa, and a flow rate of 2.5 mL/min. The resulting SPI powders were vacuum-packed and stored at −20 °C for subsequent analysis.

### 2.3. Protein Solubility

SPI powder was dissolved in deionized water at a concentration of 1% (*w*/*v*, g/mL) and magnetically stirred for 1 h at room temperature [25]. Following centrifugation (10,000× *g*, 15 min, 20 °C), the protein concentrations of both the supernatant and the SPI suspension prior to centrifugation were determined using the Biuret method [23]. Absorbance values were measured using BSA as the standard protein, and a standard curve was plotted to calculate protein concentration. Solubility was determined according to Equation (1).(1)$$Solubility\left(\%\right)=\frac{C_{1}}{C_{2}}\times 100\%$$
where *C*_1_ denotes the protein concentration of the supernatant after centrifugation, while *C*_2_ represents the protein concentration before centrifugation.

Solubility was monitored during storage at 37 °C. When the measured solubility reached 80%, 70%, 60%, and 50%, the corresponding samples were collected for the experiment. These samples were stored at −20 °C and designated as SPI80, SPI70, SPI60, and SPI50, respectively.

### 2.4. Three-Phase Contact Angle

The oil–water interfacial properties of SPIs with varying protein solubilities were assessed by measuring the three-phase contact angle using a droplet shape analyzer (CA100, Guangdong Beidou Precision Instrument Co., Ltd., Dongguan, China) [26]. SPI was first compressed into tablets with a thickness of 1–2 mm and immersed in soybean oil for 10 min. Excess oil was removed by placing the tablets on filter paper. Next, 5 μL of water was dispensed onto the tablet surface from a syringe and the image at the moment of droplet contact was captured. Images were captured using a high-speed camera, and contact angles were determined using CAPST software 2.6 (Guangdong Beidou Precision Instrument Co., Ltd., Dongguan, China). Each measurement was conducted in triplicate.

### 2.5. Hydrophobicity

The ANS hydrophobic fluorescent probe method was employed to determine the hydrophobicity of different SPI samples [27]. A 1% (*w*/*v*) SPI solution was prepared in 0.01 M phosphate buffer (pH 7.0), stirred at room temperature for 2 h, and then centrifuged at 10,000× *g* for 15 min. The protein concentration of the supernatant was determined using the Biuret method and subsequently diluted with PBS to concentrations of 0.02 mg/mL, 0.04 mg/mL, 0.06 mg/mL, 0.08 mg/mL, and 0.1 mg/mL. A 50 μL aliquot of ANS solution (8 mM) was added to 4 mL of each diluted protein solution and mixed using a vortex shaker (Vortex 1, IKA Works GmbH & Co., Kg., Baden-Württemberg, Germany). After allowing the mixture to stand for 3 min, fluorescence intensity was measured at an excitation wavelength of 365 nm and an emission wavelength of 484 nm. A linear analysis was performed by plotting fluorescence intensity against SPI protein concentration, and the obtained slope was considered an indicator of hydrophobicity.

### 2.6. Preparation of SPI MIPEs

A 1% (*w*/*v*) SPI solution was prepared by dissolving SPI in deionized water, followed by stirring for 2 h at room temperature. The solution was then stored at 4 °C overnight. Soybean oil (Φ = 60%) was added to the prepared SPI solution and homogenized for 2 min using a high-speed homogenizer (T18, IKA Works GmbH & Co., Kg., Baden-Württemberg, Germany) at 12,000 r/min to obtain SPI MIPEs.

### 2.7. Creaming Index (CI) of MIPEs

To assess the stability of SPI MIPEs, the prepared MIPE samples were sealed in glass bottles and stored at 4 °C for 15 days. Photographs were taken every 3 days to document changes during storage. The creaming Index (CI) of MIPEs was quantified based on the variation in the height of the bottom serum phase (H) over time, calculated according to Equation (2).(2)$$CI\left(\%\right)=\frac{H_{s}}{H_{t}}\times 100\%$$
where *H_s_* represents the height of the serum phase, while *H_t_* indicates the total height of all phases.

### 2.8. Particle Size Distribution of MIPEs

The particle size distribution of the prepared MIPEs was analyzed using a laser particle sizer (Nano-ZS, Malvern Instruments Ltd., Worcestershire, UK). SPI MIPEs were diluted 200-fold with a 1% (*w*/*v*) SDS solution, and the relative refractive index of the MIPEs was set to 1.095, calculated as the ratio of the refractive index of soybean oil (1.456) to that of water (1.33) [28]. The particle size distribution of SPI MIPEs was measured every 3 days over a storage period of 0–15 days.

### 2.9. ζ-Potential of MIPEs

The ζ-potential of SPI MIPEs was measured using a laser particle sizer with an immersion cell (DTS1070, Malvern Instruments Ltd., Worcestershire, UK) at 25 °C [29]. Before measurement, the MIPEs were diluted 200-fold with PBS buffer (0.01 mol/L, pH 7.0) to minimize multiple scattering effects. The ζ-potential of MIPEs was monitored every 3 days over a storage period of 0–15 days.

### 2.10. Interfacial Protein Adsorption Rate of the MIPEs

The interfacial protein adsorption of SPI MIPEs was determined following the method reported by Jiao et al. SPI MIPEs were centrifuged (H2100R, Hunan Xiangyi Laboratory Instrument Development Co., Ltd., Changsha, China) at 15,000× *g* for 45 min at room temperature. After centrifugation, the aqueous phase at the bottom was collected and filtered through a 0.45 μm filter [30]. The protein concentration of the filtrate (*C_f_*) was measured using the Lowry method [31]. Simultaneously, the initial SPI suspension used for SPI MIPE preparation was centrifuged under identical conditions, and the protein concentrations of both the supernatant (*C_s_*) and the initial SPI suspension (*C*_0_) were determined using the same method applied for *C_f_*. The interfacial protein adsorption rate was calculated according to Equation (3).(3)$$Interfacial\;protein\;adsorption\;rate\left(\%\right)=\frac{C_{f}-C_{s}}{C_{0}}\times 100\%$$
where *C_f_* represents the protein concentration of the filtrate, *C_s_* denotes the protein concentration of the supernatant, and *C*_0_ corresponds to the protein concentration of the initial SPI suspension.

### 2.11. Rheological Properties of MIPEs

To analyze the rheological properties of MIPEs stabilized with SPI of different solubility levels, dynamic viscoelastic measurements were performed using a rheometer (MCR302, Anton Paar Ltd., Graz, Austria) with a parallel plate (25 mm). An oscillatory stress sweep was conducted within a range of 0.001 to 20 Pa, followed by a dynamic frequency sweep from 0.1 to 100 Hz within the linear viscoelastic region. The storage modulus (G′) and loss modulus (G″) were obtained from the dynamic mechanical spectra as functions of oscillatory stress and frequency. Viscosity was recorded as the shear rate increased from 0.10 to 100 s^−1^. All measurements were carried out at 25 °C [32].

### 2.12. Confocal Laser Scanning Microscopy (CLSM) of MIPEs

The morphology of SPI-stabilized MIPEs with different protein solubility levels was analyzed using a laser confocal scanning microscope (LSM900, Carl Zeiss AG, Oberkochen, Germany) following the method of Ma et al. [28]. Nile Red and Nile Blue were used to stain the oil and protein phases, respectively. The excitation wavelength for Nile Red was set at 488 nm, while for Nile Blue, it was 633 nm. Image processing was performed using Zeiss Application Suite software 3.8.

### 2.13. Cryo-Scanning Electron Microscopy (Cryo-SEM) Observation of MIPEs

The morphology of SPI-stabilized MIPEs was further analyzed using a cryo-scanning electron microscope (Regulus 8100, Hitachi Ltd, Tokyo, Japan). Freshly prepared MIPEs were immediately frozen in liquid nitrogen (−196 °C) and then sublimated in a cryo-preparation apparatus at −90 °C for 20 min. After coating with a gold–palladium alloy at 10 mA for 60 s, Cryo-SEM observations were conducted at an accelerating voltage of 5.0 kV [33].

### 2.14. Statistical Analysis

All experiments were conducted in triplicate, and the results were expressed as the mean ± standard deviation. Data analysis was performed using one-way ANOVA in IBM SPSS Statistics 26.

## 3. Results

### 3.1. Interfacial Characterization of SPI

All SPI samples were obtained from a single supplier and production batch, and were processed using a standardized alkaline extraction followed by acid precipitation. Preliminary experiments revealed that SPI with approximately 40% solubility (SPI40) was unable to form stable emulsions. Consequently, SPI samples with solubility ranging from 50% to 80% were selected for use. The results of protein solubility tests for different SPIs are presented in Figure 1A. SPI80, SPI70, SPI60, and SPI50 corresponded to SPI solubilities of about 80%, 70%, 60%, and 50% (*w*/*v*), respectively.

As shown in Figure 1B, the contact angles of SPI samples ranged from 79.35° to 86.55°, with SPI60 and SPI50 exhibiting values closer to 90° compared to SPI80 and SPI70. Theoretically, when the contact angle approaches 90°, the emulsifier is more evenly distributed between the oil and water phases, potentially forming a denser particle film at the oil–water interface. This structure can effectively prevent droplet aggregation, thereby improving emulsion stability [34]. Consequently, all SPI samples demonstrated the potential to function as emulsion stabilizers, with SPI60 and SPI50 exhibiting superior characteristics.

Hydrophobic groups can strengthen the interaction between SPI and the oil phase in MIPEs, contributing to improved stabilization [35]. The H_0_ of SPI samples with different solubility levels is presented in Figure 1B. As SPI solubility decreases, H_0_ initially declines and then stabilizes, with no significant differences observed among SPI 70, SPI60, and SPI50. An increase in H_0_ is advantageous for emulsifying properties, as higher hydrophobicity more effectively reduces interfacial tension and facilitates the formation of a stable interfacial film, thereby enhancing emulsion stability [36,37]. These findings indicate that SPI with lower solubility possesses a potential for stabilizing MIPEs.

### 3.2. Storage Stability of MIPEs

The CI, as shown in Figure 2, illustrates the stability of MIPEs stabilized by SPIs with different solubility levels. The CI of SPI80 MIPEs remained nearly unchanged throughout the 15-day storage period. In contrast, SPI70 exhibited a noticeable decline on the ninth day, likely due to the onset of destabilization, potentially caused by emulsifier depletion or changes in interfacial properties over time [38]. Both SPI60 and SPI50 showed a decline from the 3rd day onward, following similar downward trends, and by the 15th day, their emulsification indices were significantly lower than that of SPI70. This may be attributed to the lower concentration of soluble proteins in SPI60 and SPI50, which might result in thinner or less robust interfacial films, making the MIPEs more susceptible to environmental stress [39].

Notably, none of the MIPEs exhibited oil precipitation, indicating low-solubility SPIs have some potential for application. Further investigation is required to elucidate the specific mechanisms and conditions responsible for the observed trends in SPI-stabilized MIPEs.

### 3.3. Zeta Potential and Particle Size

The zeta potential of MIPEs, particularly oil-in-water (O/W) MIPEs, plays a crucial role in determining their stability. As shown in Figure 3A, at the beginning of storage, SPI80 MIPEs exhibited the highest absolute zeta potential value, followed by SPI70, SPI60, and SPI50. Notably, the absolute zeta potential values of SPI80 and SPI70 exceeded 30 mV, whereas SPI50 and SPI60 were about 28 mV. It is widely recognized that MIPEs with zeta potential values above 30 mV tend to exhibit greater storage stability [40]. These findings align with the results of the CI experiment, where SPI80 MIPEs demonstrated the highest stability, followed by SPI70, while SPI50 and SPI60 showed the lowest stability.

During the 15-day storage period, a significant decline in the zeta potential of all MIPEs was observed, indicating a reduction in stability. This decline may be attributed to alterations in interfacial properties and the progressive aggregation of protein molecules over time [41]. Among the samples, SPI80 and SPI70 exhibited a slower decline, with a noticeable drop on the 15th day (*p* < 0.05), suggesting better storage stability. In contrast, SPI60 and SPI50 experienced a more rapid decrease, with a continuous and pronounced decline beginning on the sixth day (*p* < 0.05), indicating lower stability. These observations were further supported by particle size analysis.

As shown in Figure 3B–D, the particle sizes of SPI80 and SPI70 emulsions remained relatively unchanged over the 15-day storage period, whereas those of SPI60 and SPI50 increased significantly. As shown in Figure 3B and Table 1, the initial MIPEs containing SPI80 exhibited the largest particle size, about 46.2 μm, followed by SPI70 at around 41.7 μm. In contrast, SPI50 and SPI60 displayed the smallest particle sizes, about 18.1 μm and 17.3 μm, with minimal differences between them. Notably, higher protein solubility does not necessarily correlate with smaller particle sizes in MIPEs. The high solubility of SPI, such as in SPI80, may lead to increased protein adsorption at the oil-water interface, forming a more effective barrier against droplet coalescence, thereby improving MIPEs stability despite larger droplet sizes [41]. As illustrated in Figure 3B–D and Table 1, the particle size of SPI60 and SPI50 MIPEs increased over storage time. This may be attributed to the lower density of protein particles at the interface, which promotes aggregation and bonding, leading to larger particle formation and reduced MIPEs stability [42]. In contrast, the particle size of SPI80 and SPI70 MIPEs remained relatively stable, indicating better stability.

For SPI80, which exhibits the highest solubility among the tested samples, MIPE stability was maintained despite the presence of larger particle sizes. These findings indicate that the stability of MIPEs is governed more by interfacial properties than by droplet size alone. Prior research on Pickering internal phase emulsions stabilized with WPI/SPI composite particles has demonstrated high stability despite the presence of larger droplets, which has been attributed to robust interfacial interactions [39]. The mechanism by which increased SPI solubility leads to larger particle sizes while concurrently enhancing MIPEs stability remains unclear and warrants further investigation. Future studies should focus on elucidating SPI molecular interactions at the MIPEs interface and the role of viscosity, among other contributing factors.

### 3.4. Interfacial Protein Adsorption Rate of MIPEs

The interfacial protein adsorption rate is a crucial factor influencing emulsion stability. As shown in Figure 4, a significant decrease (*p* < 0.05) in the interfacial protein adsorption rate of MIPEs was observed as SPI solubility declined. This trend aligns with the results of the CI (Figure 2). Generally, a higher interfacial protein adsorption rate contributes to greater MIPE stability by forming a stronger protective layer at the oil–water interface [43].

Furthermore, as noted earlier, SPI80 produced the most stable MIPEs despite having the largest particle size. This outcome is likely due to its high solubility and the associated elevated rate of protein adsorption at the interface, which may offset the commonly observed relationship between smaller particle sizes and improved emulsion stability. Delahaije et al. demonstrated that emulsions were stabilized against flocculation if the protein concentration was high enough to completely cover the interface [44].

### 3.5. Rheological Behavior of MIPEs

Figure 5 presents the rheological behavior of MIPEs stabilized with SPI at different solubility levels. The storage modulus (G′) and loss modulus (G″) as functions of oscillation stress are shown in Figure 5A. Within the linear viscoelastic region, G′ remains dominant over G″ and remains nearly constant as stress increases. A similar trend was observed in the frequency sweep curves (Figure 5B). These behaviors are characteristic of a weak gel network, which is associated with the formation of a three-dimensional structure through droplet interactions within MIPEs. When the applied stress exceeded the linear viscoelastic region, both G′ and G″ decreased sharply in all MIPEs, and a yield stress was observed, identified as the crossover point of G′ and G″. This transition typically signifies structural breakdown and network transformation. As shown in Figure 5A, SPI80 MIPEs exhibit the highest yield stress, and the overall trend indicates a positive correlation between solubility and stress, which further validates the trend observed in the CI (Figure 2).

As SPI solubility decreased, the overall moduli (G′ and G″) within both stress (Figure 5A) and frequency (Figure 5B) scan ranges also declined, indicating a weakening of the MIPE network structure and a possible transition toward instability. This reduction in structural integrity may be attributed to the lower protein adsorption rate at the oil droplet interface (Figure 4), suggesting that decreased SPI solubility in MIPEs weakens droplet interactions and compromises overall emulsion stability.

The viscosity of MIPEs decreased with increasing shear rate, exhibiting typical shear-thinning behavior. This phenomenon can be attributed to the aggregation and coalescence of oil droplets as shear rate increases. Additionally, viscosity was found to decrease with the reduction in SPI solubility, which may be due to the more dispersed arrangement of oil droplets within the MIPEs and the formation of a weaker gel network structure. Yang et al. demonstrated that the elevated viscosity originates from strong intermolecular interactions in the aqueous phase, driven by high interfacial protein adsorption rates [45]. Notably, the viscosity trend of MIPEs aligns with their CI variations (Figure 2), where higher viscosity corresponds to enhanced stability. This phenomenon can be attributed to the retarded droplet sedimentation and reduced droplet–droplet interactions under high-viscosity conditions, which collectively improve emulsion stabilization. Among the samples, SPI50 MIPEs, with the lowest solubility, exhibited a rougher surface and a higher presence of free protein particles between droplets, contributing to lower viscosity, as observed in the CLSM images (Figure 6).

### 3.6. Microstructure of MIPEs

To further analyze the distribution and interfacial morphology of SPI-stabilized MIPEs, the microstructure of the MIPEs was examined using CLSM and Cryo-SEM. As shown in Figure 6A–C, (A) represents Nile Blue-stained proteins, (B) displays Nile Red-stained oil combined with Nile Blue-stained proteins, and (C) shows Nile Red-stained oil. The SPI80 MIPEs exhibited the largest particle size, while SPI60 MIPEs and SPI50 MIPEs displayed relatively smaller particles, consistent with the particle size measurements (Figure 3B). Furthermore, SPI nanoparticles—likely representing insoluble components—were observed in both the continuous phase and at droplet interfaces. This observation suggests that insoluble particles participate in the formation of interfacial films; however, their weak intermolecular interactions result in reduced interfacial protein adsorption rates (Figure 4), ultimately leading to diminished stability of the MIPEs (Figure 2).

As shown in Figure 6A–C, MIPEs stabilized with SPI80 exhibited a HIPE-like compact structure, where oil droplets (red) were relatively evenly dispersed and encapsulated by a protein layer (green) adsorbed at the O/W interface. With decreasing SPI solubility, as observed in MIPEs stabilized with SPI70, SPI60, and SPI50, the microstructure became increasingly irregular, with larger interstitial spaces leading to a less compact structure. This non-uniform dispersion of oil droplets suggests the occurrence of droplet coalescence. Moreover, the proportion of the continuous phase increased, with a greater amount of protein distributed within it, which aligns with the previously observed reduction in interfacial protein adsorption (Figure 4).

The Cryo-SEM results further illustrate the microstructure of the MIPEs, revealing numerous round or ellipsoidal spheres in each sample, confirming the formation of oil droplets encapsulated by SPI particles. The observed droplet size trends (Figure 3) are consistent with the CLSM images and particle size measurements. In MIPEs stabilized with SPI80, larger droplets were consistently present, and these larger droplets exhibited enhanced stability. In contrast, smaller droplets were more prone to aggregation to some extent. The microstructure of SPI80 MIPEs exhibited a relatively smooth interface, with the surface film displaying a multilayered wrapping morphology (Figure 6E). This observation indicates that the interfacial film of SPI80 MIPEs possesses a certain thickness, and the higher distribution of SPI at the interface correlates with elevated interfacial protein adsorption rates (Figure 4). Notably, the interfacial protein films in SPI80 MIPEs were interconnected through intermolecular cross-linking, maintaining stable interactions. In contrast, SPI50-stabilized MIPEs showed rougher surface textures, with no observable cross-linking between films, suggesting weakened intermolecular interactions. This phenomenon is likely attributed to the diminished solubility of SPI50, which led to reduced rheological performance and smaller yield stress (Figure 5A), ultimately compromising interfacial stability.

An intriguing observation in this study was that lower SPI solubility led to the formation of smaller emulsion droplets, yet resulted in reduced emulsion stability. Emulsion stability is governed by multiple factors, including but not limited to droplet size. The findings indicate that decreased SPI solubility corresponds with lower absolute zeta potential, reduced interfacial protein adsorption, diminished viscoelastic properties, and decreased viscosity in the resulting SPI-MIPEs. These results suggest that the stability of SPI-HIPEs is primarily governed by the zeta potential, the characteristics of the interfacial protein film, and the rheological behavior of the system, rather than droplet size alone. Microscopic analysis further confirmed that higher SPI solubility leads to the formation of a thicker interfacial film and promotes stronger particle–particle interactions.

## 4. Conclusions

In conclusion, despite the decline in solubility, SPI retains its potential to stabilize emulsions, though with varying effectiveness. The reduction in solubility significantly increased the contact angle and surface hydrophobicity of SPI (*p* < 0.05), suggesting that low-solubility SPIs can serve as effective stabilizers for emulsions. While all SPI samples facilitated the formation of MIPEs, reduced SPI solubility adversely affected the storage stability of the emulsions. The zeta potential and viscosity measurements indicated that this instability may be attributed to the lower zeta potential and weaker viscosity properties of MIPEs stabilized by low-solubility SPI, such as SPI60 and SPI50. Interestingly, lower SPI solubility resulted in smaller droplet sizes in MIPEs but was associated with a lower CI during storage. CLSM and Cryo-SEM analyses revealed that this phenomenon was primarily due to weaker molecular interactions and the formation of a less compact gel network structure in MIPEs stabilized by lower-solubility SPI. However, no oil separation was observed on the surface of MIPEs within 15 days of storage, and the reduction in CI remained below 20%. These results suggest that system stability may be further enhanced by optimizing factors such as Zeta potential, viscosity, and interfacial film thickness. The findings offer a theoretical foundation for the use of commercially available SPI, with a solubility range of 50–60%, in MIPE systems, supporting its potential application in plant-based dairy alternatives.

## Figures and Tables

**Figure 1 foods-14-02028-f001:**
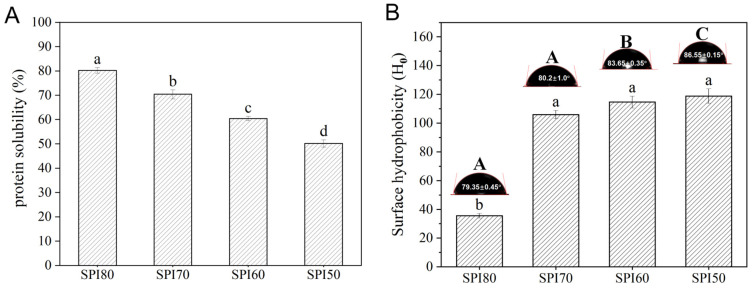
Protein solubility variations (**A**), surface hydrophobicity (H_0_), and three-phase contact angles of SPI80, SPI70, SPI60, and SPI50 (**B**) (different letters in the figure indicate statistically significant differences, *p* < 0.05. Uppercase letters (A, B, C) denote significant differences in contact angle values, while lowercase letters (a, b) indicate significant differences in hydrophobicity).

**Figure 2 foods-14-02028-f002:**
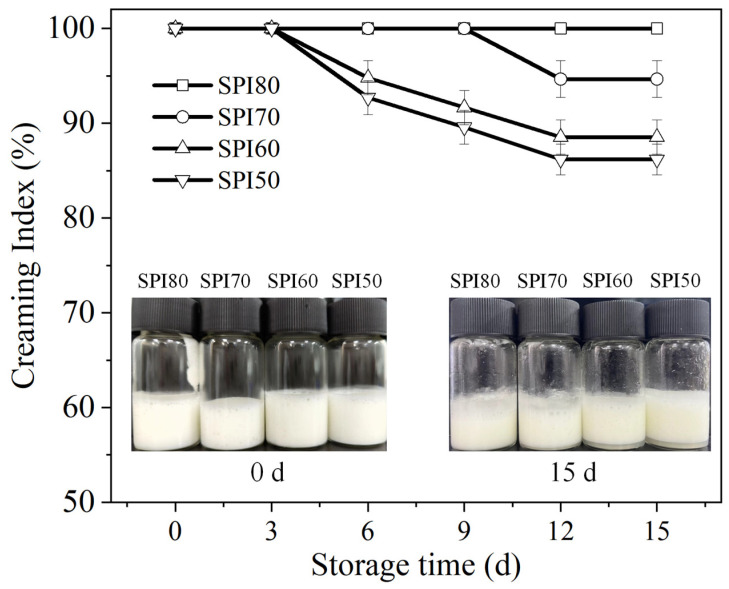
Creaming Index (CI) of medium internal phase emulsions (MIPEs) stabilized by SPI with different solubility levels during storage at 4 °C, where SPI80, SPI70, SPI60, and SPI50 correspond to SPI solubilities of about 80%, 70%, 60%, and 50% (*w*/*v*), respectively.

**Figure 3 foods-14-02028-f003:**
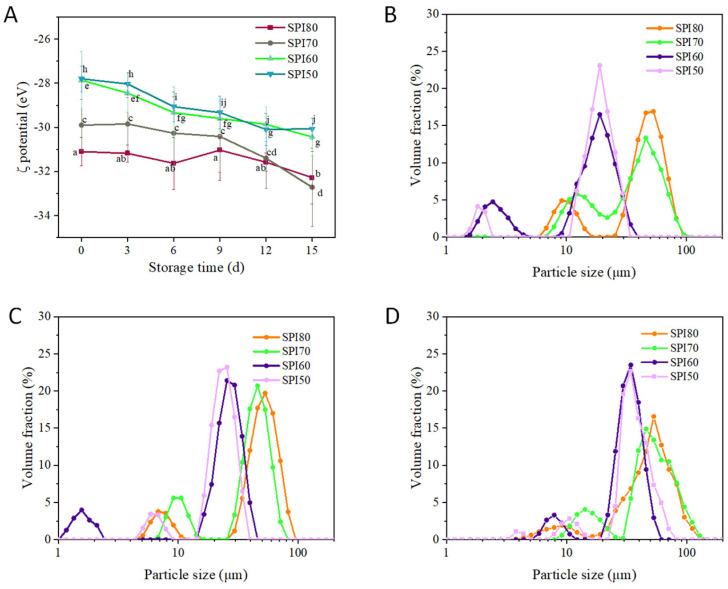
(**A**) Zeta potential of medium internal phase emulsions (MIPEs) stabilized by SPI during storage for 15 days at 4 °C; (**B**–**D**) particle size distribution from 0 to 15 days and changes in particle size of MIPEs at 0 days (**B**), 6 days (**C**), and 15 days (**D**). SPI80, SPI70, SPI60, and SPI50 correspond to SPI solubilities of about 80%, 70%, 60%, and 50% (*w*/*v*), respectively (different letters in the figure indicate statistically significant differences, *p* < 0.05).

**Figure 4 foods-14-02028-f004:**
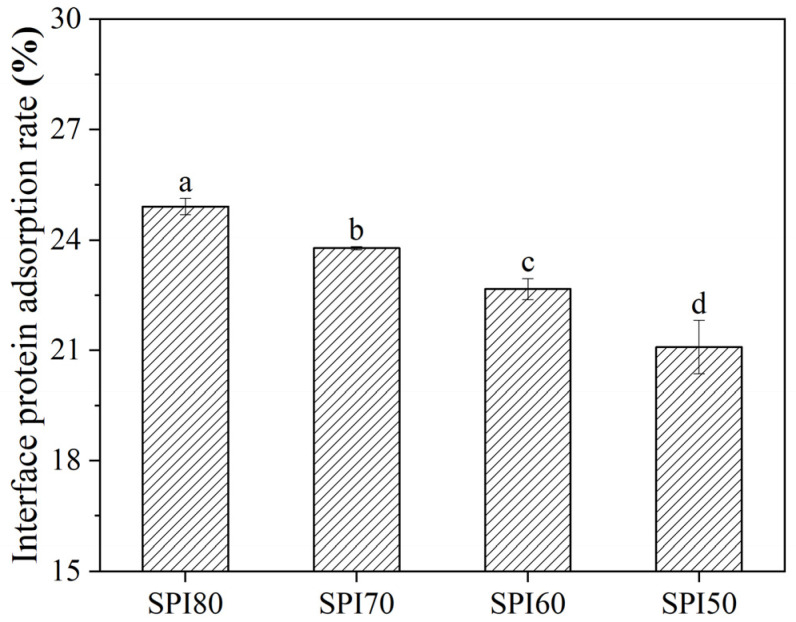
Interfacial protein adsorption rate of medium internal phase emulsions (MIPEs) stabilized by SPI with different solubility levels, where SPI80, SPI70, SPI60, and SPI50 correspond to SPI solubilities of about 80%, 70%, 60%, and 50% (*w*/*v*), respectively (different letters in the figure indicate statistically significant differences, *p* < 0.05).

**Figure 5 foods-14-02028-f005:**
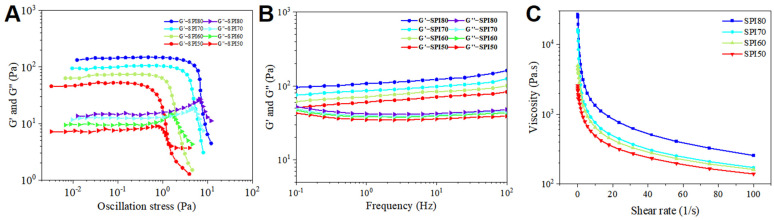
Rheological behavior of medium internal phase emulsions (MIPEs) stabilized by SPI at 25 °C: (**A**) storage modulus (G′, Pa) and loss modulus (G″, Pa) of MIPEs as a function of oscillation stress (Pa); (**B**) G′ and G″ of MIPEs as a function of frequency (Hz); (**C**) viscosity (Pa·s) of MIPEs stabilized by SPI as a function of shear rate. SPI80, SPI70, SPI60, and SPI50 correspond to SPI solubilities of about 80%, 70%, 60%, and 50% (*w*/*v*), respectively.

**Figure 6 foods-14-02028-f006:**
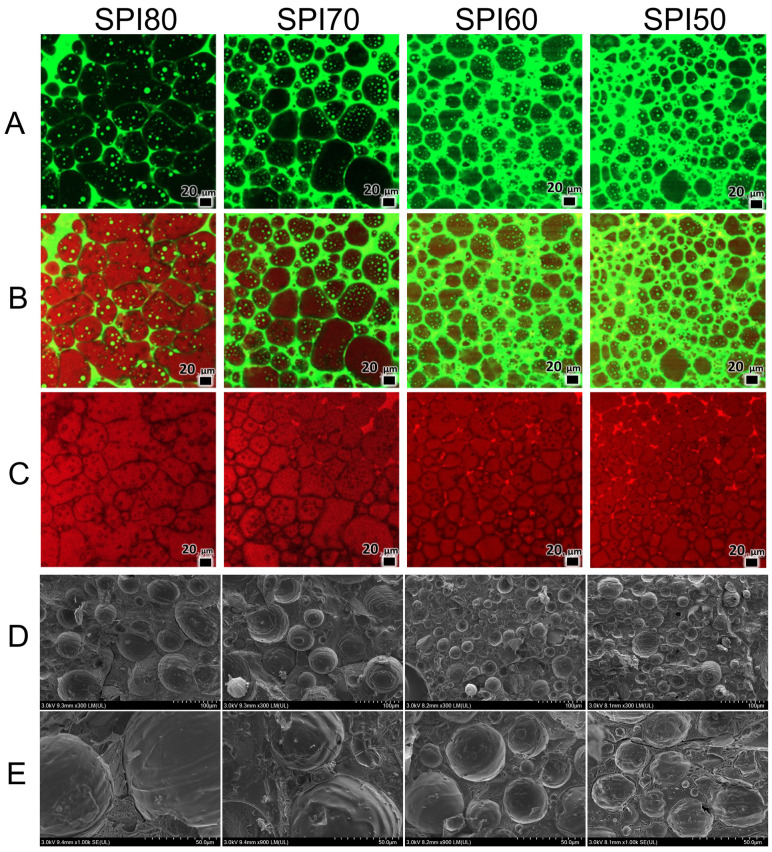
Morphology of medium internal phase emulsions (MIPEs) stabilized by SPI at 25 °C: (**A**–**C**) confocal laser scanning microscopy (CLSM) images stained with Nile Blue (**A**), Nile Red (**C**), and a combination of Nile Blue and Nile Red (**B**); (**D**,**E**) Cryo-SEM images of MIPEs with scales of 100 μm (**D**) and 50 μm (**E**), respectively. SPI80, SPI70, SPI60, and SPI50 correspond to SPI solubilities of about 80%, 70%, 60%, and 50% (*w*/*v*), respectively.

**Table 1 foods-14-02028-t001:** Particle size (D_4,3_) of medium internal phase emulsions (MIPEs) stabilized by SPI during storage for 15 days at 4 °C (Within the same storage time, different lowercase letters (a, b) indicate statistically significant differences in particle size between treatments, *p* < 0.05).

Storage Time	D_4,3_ (μm)			
	SPI80	SPI70	SPI60	SPI50
0 days	46.2 ± 6.8 ^a^	41.7 ± 3.6 ^a^	17.3 ± 1.5 ^b^	18.1 ± 1.5 ^b^
6 days	48.3 ± 4.2 ^a^	43.8 ± 3.6 ^a^	24.3 ± 2.0 ^b^	23.2 ± 2.0 ^b^
15 days	50.7 ± 4.2 ^a^	48.3 ± 4.2 ^a^	37.5 ± 2.9 ^b^	35.9 ± 2.9 ^b^

## Data Availability

The original contributions presented in this study are included in the article. Further inquiries can be directed to the corresponding author.

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
