# Peer review of "Medium Internal Phase Emulsions Stabilized by Soy Protein Isolates: Protein Solubility Effect and Stabilization Mechanism"

_foods, 2025, doi:10.3390/foods14122028_

Round 1
Reviewer 1 Report
Comments and Suggestions for Authors
The manuscript titled “Medium internal phase emulsions stabilized by soy protein isolates: protein solubility effect and stabilization mechanism” by Fengxian Guo et al. provide a study examines the effect of soybean protein isolate (SPI) solubility on the formation and stability of medium internal phase emulsions (MIPEs). SPI variants with solubilities of ~80% to ~50% (SPI80 to SPI50) were used.
The topic covered in the article is highly relevant and would be of significant interest to readers, making it suitable for publication. The manuscript is well written, organized, and it is suitable for publication after minor revisions.
- Could the authors consider presenting the results shown in Figure 3 in tabular form as well, to clearly illustrate the changes in particle size distribution over time? The current graphical representation makes it difficult to accurately assess the numerical evolution of particle size. Including a table with key values would improve clarity and facilitate comparison, thereby enhancing the scientific rigor and transparency of the data presentation.
- It is essential that the authors provide comprehensive information regarding the statistical analysis conducted, particularly the ANOVA, in Section 3 (Results). The manuscript should include a clear description of the statistical methodology, as well as a detailed interpretation of the ANOVA results. This information is necessary to validate the significance of the reported differences and to ensure the scientific rigor of the study.
- The authors should specify the time point at which the contact angle shown in Figure 1b was measured. Was it recorded immediately after the water droplet was placed on the surface, or after a certain period of time? This clarification is important for the proper interpretation of the wettability results.
- Do the authors anticipate improved results (such as more stable emulsions or larger particle sizes compared to those obtained with SPI80) if experiments were conducted using SPI90, corresponding to a solubility of approximately 90%? A brief discussion of this possibility would enhance the interpretation of the findings.
Author Response
Reviewer #1: The manuscript titled “Medium internal phase emulsions stabilized by soy protein isolates: protein solubility effect and stabilization mechanism” by Fengxian Guo et al. provide a study examines the effect of soybean protein isolate (SPI) solubility on the formation and stability of medium internal phase emulsions (MIPEs). SPI variants with solubilities of ~80% to ~50% (SPI80 to SPI50) were used.
The topic covered in the article is highly relevant and would be of significant interest to readers, making it suitable for publication. The manuscript is well written, organized, and it is suitable for publication after minor revisions.
Comments 1. Could the authors consider presenting the results shown in Figure 3 in tabular form as well, to clearly illustrate the changes in particle size distribution over time? The current graphical representation makes it difficult to accurately assess the numerical evolution of particle size. Including a table with key values would improve clarity and facilitate comparison, thereby enhancing the scientific rigor and transparency of the data presentation.
Response 1: Thank you for your insightful suggestion. We agree that presenting the particle size distribution data in tabular form alongside the graphical representation will significantly enhance the clarity and accessibility of the results. In response, we have added Table 1 to the revised manuscript, which presents the key numerical values corresponding to Figure 3, including D4,3 (volume-weighted mean diameter).
Comments 2. It is essential that the authors provide comprehensive information regarding the statistical analysis conducted, particularly the ANOVA, in Section 3 (Results). The manuscript should include a clear description of the statistical methodology, as well as a detailed interpretation of the ANOVA results. This information is necessary to validate the significance of the reported differences and to ensure the scientific rigor of the study.
Response 2: We greatly appreciate your valuable feedback. We fully agree that a clear description and proper interpretation of statistical analysis are crucial for ensuring the scientific rigor of our study. In Section 3.3, we have expanded our interpretation of the ANOVA results. Additionally, we have updated Figure 3A to include statistical significance indicators (e.g., letters a–j) to visually represent the ANOVA results. This improvement enhances the transparency and interpretability of the data.
Comments 3. The authors should specify the time point at which the contact angle shown in Figure 1b was measured. Was it recorded immediately after the water droplet was placed on the surface, or after a certain period of time? This clarification is important for the proper interpretation of the wettability results.
Response 3: Thank you for your reminding. We have revised them and changes are indicated in red fonts in the main text.
Page 3, Line 132-133: Next, 5 μL of water was dispensed onto the tablet surface from a syringe and the image at the moment of droplet contact was captured.
Comments 4. Do the authors anticipate improved results(such as more stable emulsions or larger particle sizes compared to those obtained with SPI80) if experiments were conducted using SPI90, corresponding to a solubility of approximately 90%? A brief discussion of this possibility would enhance the interpretation of the findings.
Response 4: We thank the reviewer for raising this insightful question regarding the potential impact of using SPI with higher solubility (e.g., SPI90) on emulsion properties. We agree that exploring SPI solubility closer to its theoretical maximum is an interesting theoretical possibility. However, based on our specific extraction and processing conditions, we consistently observed that the maximum solubility achievable for the SPI used in this study was approximately 80% (SPI80). we were unable to attain SPI fractions with solubility reliably reaching 90% (SPI90) under the conditions employed.
In fact, many literature reports indicate that SPI typically exhibits a solubility of approximately 80% [1-4]. Meanwhile, China's GB 20371-2016 National Food Safety Standard for Food Use Vegetable Protein stipulates that the nitrogen solubility index (NSI) of commercially produced SPI must exceed 80% at the time of factory release. However, during storage deterioration, the solubility of commercial SPI often drops significantly below 80% [5].
[1]Huang, L., Zhang, W., Yan, D., Ma, L., Ma, H. . Solubility and aggregation of soy protein isolate induced by different ionic liquids with the assistance of ultrasound. International Journal of Biological Macromolecules, 2020, 164, 2277-2283.
[2]Tiong, A. Y., Crawford, S., Jones, N. C., McKinley, G. H., Batchelor, W., van’t Hag, L. (2024). Pea and soy protein isolate fractal gels: The role of protein composition, structure and solubility on their gelation behaviour. Food Structure, 2024, 40, 100374.
[3]Liu, D., Zhang, L., Wang, Y., Li, Z., Wang, Z., Han, J. Effect of high hydrostatic pressure on solubility and conformation changes of soybean protein isolate glycated with flaxseed gum. Food Chemistry, 2020, 333, 127530.
[4]Wang, N., Zhou, X., Wang, W., Wang, L., Jiang, L., Liu, T., Yu, D. Effect of high intensity ultrasound on the structure and solubility of soy protein isolate-pectin complex. Ultrasonics Sonochemistry, 2021, 80, 105808.
[5]Schmid, E. M., Farahnaky, A., Adhikari, B., Savadkoohi, S., Torley, P. J. Investigation into the physiochemical properties of soy protein isolate and concentrate powders from different manufacturers. International Journal of Food Science and Technology, 2024, 59, 1679–1693.
Reviewer 2 Report
Comments and Suggestions for Authors
A few recommendations for the authros:
The reviewer believes that this is a well formulated manuscript exploring on a very interesting topic of both scientific and commercial interest such as protein solubility effect and stabilization mechanism in medium internal phase emulsions stabilized by soy protein isolates.
Overall, the authors present their research findings in a clear and well comprehensive manner by using an exhaustive amount of updated literature evidence in this scientific field. A couple of comments /recommendations are given below for the consideration of the authors:
- Figures 1 and 4. Please present the order of activity in terms of statistical differences (e.g. a>b>c…)
- According to the results, reduced SPI solubility adversely affected the storage stability of the emulsions…Is there any solubility threshold below which destbilisation observed in these emulsion systems? (if not explored in this work, any threshold reported in the literature?)
- The CSLM/SEM images of these experiments show that with decreasing SPI solubility, the emulsions microstructure becomes increasingly irregular, with larger interstitial spaces leading to a less compact structure. If I get it right SP180 emulsions (with larger droplet sizes) are therefore microstructurally more stable than SP150 stabilised emulsions. This may sound contradictory to studies in other protein stabilised emulsions (e.g. my experience on WP/caseinate emulsions) where smaller droplet sizes lead to physically more stable emulsions products (e.g non-dairy type spreads…)…Can you please elaborate on this via a dedicated paragraph at the end of section 3?
- It would be easier for the reader if you could listin bullet points the most important findings of this work (e.g. highlighting the challenges and any potential for commercial applications). Could you please refer examples of real products on which the findings of these emulsion systems could have a direct applications?
Author Response
Reviewer #2: The reviewer believes that this is a well formulated manuscript exploring on a very interesting topic of both scientific and commercial interest such as protein solubility effect and stabilization mechanism in medium internal phase emulsions stabilized by soy protein isolates.
Overall, the authors present their research findings in a clear and well comprehensive manner by using an exhaustive amount of updated literature evidence in this scientific field. A couple of comments /recommendations are given below for the consideration of the authors:
Comments 1. Figures 1 and 4. Please present the order of activity in terms of statistical differences (e.g. a>b>c…)According to the results, reduced SPI solubility adversely affected the storage stability of the emulsions…Is there any solubility threshold below which destbilisation observed in these emulsion systems? (if not explored in this work, any threshold reported in the literature?)
Response 1: Thank you very much for this insightful and constructive comment. We have already used SPI with 40% solubility in preliminary experiments, but it failed to stabilize the emulsion, so we did not proceed with further experiments. We have now included this discussion in the revised manuscript (Discussion section) to clarify the role of solubility and its lower functional limit.
Page 5, Line 220-224: Preliminary experiments revealed that SPI with approximately 40% solubility (SPI40) was unable to form stable emulsions. Consequently, SPI samples with solubility ranging from 50% to 80% were selected for use.
Comments 2. The CLSM/SEM images of these experiments show that with decreasing SPI solubility, the emulsions microstructure becomes increasingly irregular, with larger interstitial spaces leading to a less compact structure. If I get it right SP180 emulsions (with larger droplet sizes) are therefore microstructurally more stable than SP150 stabilised emulsions. This may sound contradictory to studies in other protein stabilised emulsions (e.g. my experience on WP/caseinate emulsions) where smaller droplet sizes lead to physically more stable emulsions products (e.g non-dairy type spreads…)…Can you please elaborate on this via a dedicated paragraph at the end of section 3?
Response 2: We are very grateful for your suggestion and have added a paragraph at the end of Section 3 (Page 11, Line 416-426: An intriguing observation in this study was that lower SPI solubility led to the formation of smaller emulsion droplets, yet resulted in reduced emulsion stability. Emulsion stability is governed by multiple factors, including but not limited to droplet size. The findings indicate that decreased SPI solubility corresponds with lower absolute zeta potential, reduced interfacial protein adsorption, diminished viscoelastic properties, and decreased viscosity in the resulting SPI-MIPEs. These results suggest that the stability of SPI-HIPEs is primarily governed by the zeta potential, the characteristics of the interfacial protein film, and the rheological behavior of the system, rather than droplet size alone. Microscopic analysis further confirmed that higher SPI solubility leads to the formation of a thicker interfacial film and promotes stronger particle – particle interactions.). Additionally, we have revised some earlier sections in the manuscript to ensure a clearer presentation of this concept and to align with the new explanation provided in the added paragraph. We hope this revision adequately clarifies your concern and improves the overall clarity of the manuscript. Thank you again for your valuable feedback.
Page 8, Line 300-308: These findings indicate that the stability of MIPEs is governed more by interfacial properties than by droplet size alone. Prior research on Pickering internal phase emulsions stabilized with WPI/SPI composite particles has demonstrated high stability despite the presence of larger droplets, which has been attributed to robust interfacial interactions [39]. The mechanism by which increased SPI solubility leads to larger particle sizes while concurrently enhancing MIPEs stability remains unclear and warrants further investigation. Future studies should focus on elucidating SPI molecular interactions at the MIPEs interface and the role of viscosity, among other contributing factors.
Page 9, Line 325-329: Furthermore, as noted earlier, SPI80 produced the most stable MIPEs despite having the largest particle size. This outcome is likely due to its high solubility and the associated elevated rate of protein adsorption at the interface, which may offset the commonly observed relationship between smaller particle sizes and improved emulsion stability.
Comments 3. It would be easier for the reader if you could listin bullet points the most important findings of this work (e.g. highlighting the challenges and any potential for commercial applications). Could you please refer examples of real products on which the findings of these emulsion systems could have a direct applications?
Response 3: We sincerely appreciate your valuable suggestion. In the revised manuscript, we have added a concise summary of the key findings at the end of the Discussion section to enhance clarity and accessibility for readers. Additionally, we now refer to real-world product examples where these SPI-based emulsion systems may find direct or potential application.We believe this addition improves the practical relevance and readability of the manuscript. Thank you again for your thoughtful recommendation.
Page 13, Line 447-452: However, within 15 days of storage, no oil separation was observed on the surface of MIPEs, and the decrease in EI remained within 20%. Based on these results, it is concluded that further improvements in system stability can be achieved by optimizing parameters such as Zeta potential, viscosity, and interfacial film thickness. These findings provide a theoretical basis for the application of commercially available SPI with a solubility of 50%–60% in MIPEs systems, enabling its use in plant-based dairy alternatives.
Reviewer 3 Report
Comments and Suggestions for Authors
Dear Authors, I have some suggestions for your paper.
Please specify more about author contributions, they seem too simplistic and overlapping, for example, supervision and visualization seem like the same thing, and funding acquisition alone is not enough for one person. So, a more precise description would help.
Please provide original resolution images that are used for Figure 6.
Figure 1. requires a detailed explanation of what is shown.
Figure 4. resolution should not be blurry (decrease size?)
Author Response
Reviewer #3: Dear Authors, I have some suggestions for your paper.
Comments 1. Please specify more about author contributions, they seem too simplistic and overlapping, for example, supervision and visualization seem like the same thing, and funding acquisition alone is not enough for one person. So, a more precise description would help.
Response 1: Thank you for your reminding. We have revised them and the changes are indicated in red fonts in the main text (Page 13, Line 454-459: Conceptualization, Z.Z. and F.G.; methodology, F.G., Y.M., Y.C. and Shiying Wu; validation, H.C.; formal analysis, F.G.; investigation, B.W. and Shunhong Wu; resources, Z.Z.; data curation, Y.M. and Y.C.; writing—original draft preparation, F.G.; writing—review and editing, F.G., Y.M. and Z.Z.; visualization, Z.H.; supervision, Z.Z.; project administration, Z.Z.; funding acquisition, F.G. and Z.Z. All authors have read and agreed to the published version of the manuscript.).
Comments 2. Please provide original resolution images that are used for Figure 6.
Response 2: Thank you for your valuable suggestion. We have provided the original resolution images used for Figure 6. as supplementary files along with this revised manuscript. These high-resolution images ensure clearer visualization of the microstructure details and support the conclusions drawn in the manuscript. Please let us know if further modifications or specific formats are required.
Comments 3. Figure 1. requires a detailed explanation of what is shown.
Response 3: Thank you for your insightful comment. We have revised the figure legend and corresponding description in the manuscript to provide a more detailed explanation of Figure 1.
Page 5, Line 220: All SPI samples were sourced from the same supplier and production batch, and processed using a uniform alkaline extraction and acid precipitation method. In our preliminary experiments, SPI with solubility around 40% (SPI40) failed to form stable emulsions. Therefore, SPI with a solubility of 50%-80% was used for further study. The results of protein solubility tests for different SPIs are presented in Fig. 1A. SPI80, SPI70, SPI60, and SPI50 corresponded to SPI solubilities of about 80%, 70%, 60%, and 50% (w/v), respectively.
Comments 4. Figure 4. resolution should not be blurry (decrease size?)
Response 4: Thank you for pointing this out. We have carefully reviewed Figure 4. and agree that the image quality can be improved. In the revised manuscript, we have replaced Figure 4. with a higher-resolution version to ensure better clarity. Additionally, we have adjusted its display size to optimize sharpness without compromising readability. We hope the updated figure meets your expectations. Please let us know if further adjustment is needed.